# Clinical Factors and Biomarkers Associated with Depressive Disorders in Older Patients Affected by Chronic Kidney Disease (CKD): Does the Advanced Glycation End Products (AGEs)/RAGE (Receptor for AGEs) System Play Any Role?

**DOI:** 10.3390/geriatrics9040099

**Published:** 2024-07-30

**Authors:** Massimiliano Buoli, Elena Dozio, Lara Caldiroli, Silvia Armelloni, Elena Vianello, Massimiliano Corsi Romanelli, Giuseppe Castellano, Simone Vettoretti

**Affiliations:** 1Department of Neurosciences and Mental Health, Fondazione IRCCS Ca’ Granda Ospedale Maggiore Policlinico, 20122 Milan, Italy; 2Department of Pathophysiology and Transplantation, Università degli Studi di Milano, 20122 Milan, Italy; 3Department of Biomedical Science for Health, Università degli Studi di Milano, 20133 Milan, Italy; elena.dozio@unimi.it (E.D.); elena.vianello@unimi.it (E.V.); mmcorsi@unimi.it (M.C.R.); 4Experimental Laboratory for Research on Organ Damage Biomarkers, IRCCS Istituto Auxologico Italiano, 20149 Milan, Italy; 5Unit of Nephrology Dialysis and Kidney Transplantation, Fondazione IRCCS Ca’ Granda Ospedale Maggiore Policlinico, 20122 Milan, Italy; lara.caldiroli@policlinico.mi.it (L.C.); silvia.armelloni@policlinico.mi.it (S.A.); giuseppe.castellano@unimi.it (G.C.); simone.vettoretti@policlinico.mi.it (S.V.); 6Department of Experimental and Clinical Pathology, IRCCS Istituto Auxologico Italiano, 20149 Milan, Italy; 7Department of Clinical Sciences and Community Health, Università degli Studi di Milano, 20122 Milan, Italy

**Keywords:** chronic kidney disease, depression, advanced glycation end products, receptor for advanced glycation end products, inflammation, oxidative stress

## Abstract

Depressive disorders are highly prevalent among subjects suffering from chronic kidney disease (CKD). The aim of the present study is to evaluate clinical and biochemical factors associated with depressive disorders in a sample of older CKD patients, with a focus on advanced glycation end products (AGEs) and their soluble receptors (sRAGEs). A total of 115 older subjects affected by CKD (stages 3 to 5, not in dialysis) were selected for this study. These patients were divided into two groups according to the presence of depressive disorders defined by a score ≥ 10 on the 30-item Geriatric Depression Scale (GDS). The two groups were compared by independent sample t tests for continuous variables and χ^2^ tests for qualitative ones. Significant variables at univariate analyses were then inserted as predictors of a binary logistic regression model, with the presence or absence of depressive disorders as a dependent variable. The binary logistic regression model showed that patients with concomitant depressive disorders were more frequently of female gender (*p* < 0.01) and had lower MCP1 (*p* < 0.01) and AGE circulating levels (*p* < 0.01) than their counterparts. Depressive disorders in older CKD patients are more prevalent in women and seem to be inversely associated with systemic inflammation and circulating AGEs.

## 1. Introduction

Depressive disorders, including major depressive disorder (MDD), represent prevalent and disabling conditions that are often concomitant with a number of medical conditions, such as autoimmune diseases (e.g., rheumatoid arthritis and atopic dermatitis) [1,2], cardiovascular conditions [3], overweight [4,5], and chronic kidney disease (CKD) [6]. CKD, in turn, was identified as a leading public health problem worldwide, considering that the global prevalence of this condition was estimated to correspond to 13.4% [7]. Similarly to what happens with other medical conditions, depressive disorders may complicate the CKD course, affecting patients’ treatment compliance and quality of diet [8]. On the other hand, risk factors for the development of depressive disorders in patients affected by CKD include the severity of renal dysfunction, female gender, comorbid cardiovascular diseases, and impaired physical ability [9,10].

Both MDD [11] and CKD [12] are characterized by abnormalities in different biological systems that are responsible for alterations in a number of detectable peripheral biomarkers. Of note, patients affected by CKD show endothelial dysfunction [13], over-inflammation [14], impairment of anti-oxidant systems [15], and metabolism dysregulation [16]. Similarly, the presence and severity of depressive disorders were associated with over-activation of innate and cell-mediated immunity [17], hormonal alterations, especially in the hypothalamus–pituitary–adrenal (HPA) axis [18], hypercholesterolemia [19], and multi-vitamin deficiencies [20]. Of note, previous research by our group demonstrated that more severe depressive symptoms in patients suffering from CKD were associated with higher plasma levels of the active heterodimer of interleukin-12 (IL-12p70) [21] and low-density lipoprotein (LDL) [22].

Advanced glycation end products (AGEs) are a heterogeneous family of compounds that can be found in food or are produced in vivo because of metabolic processes. In general, they are produced by the non-enzymatic reaction of reducing sugars, like glucose and fructose, with free amine groups of proteins, lipids, and proteins. These first glycated products can then undergo further rearrangement, yielding irreversible molecules [23]. This family of compounds is receiving increasing interest because they promote the oxidative state and inflammation by interacting with a cell membrane receptor (RAGE), which is ubiquitously expressed [24]. A RAGE can also exist as a soluble molecule (sRAGE). sRAGEs are composed of two different forms: endogenous secretory RAGEs (esRAGEs), produced by alternative splicing, and cleaved RAGEs (cRAGEs), derived from the cleavage of membrane RAGEs. Differently from the membrane form, an sRAGE works as a decoy receptor. By binding circulating AGEs, it blocks RAGE activation and RAGE-related responses [25]. Along with circulating AGEs, it has been studied as a potential biomarker in many diseases [26,27,28].

Recent research highlighted that in patients with CKD, (1) sRAGE isoforms are associated with malnutrition [29] and (2) AGEs are involved in illness progression and complications [30]. A negative effect of AGEs was also reported for mental disorders [31]. In particular, preliminary findings indicated that the accumulation of these products is involved in the development of depressive disorders [32].

Considering the negative impact that depressive symptoms have on the prognosis of CKD patients, the aim of this study was to identify which clinical factors and biomarkers may be associated with the presence of depression in a sample of older CKD patients. In particular, we focused our research on the possible association between AGEs, sRAGE forms, and depressive symptoms in these patients.

## 2. Materials and Methods

### 2.1. Study Design and Patient Selection

We evaluated cross-sectionally 115 CKD-prevalent patients attending our outpatient clinic between September 2016 and March 2018. Included subjects were affected by stable CKD (stages 3 to 5, not in dialysis). eGFR was estimated by the modified CKD-EPI (Chronic Kidney Disease Epidemiology Collaboration) formula as described by Skali and co-workers [33], and clinical stability was defined as an eGFR decline ≤3 mL/min in the previous 12 months. Exclusion criteria consisted of medical conditions that may directly influence systemic inflammation and oxidative stress, such as infectious diseases in the previous month, solid tumors or hematological diseases, advanced cirrhosis, heart failure (NYHA classes 3 and 4), nephritic syndrome, inflammatory bowel diseases, and subjects under treatment with immunosuppressive drugs. We also excluded subjects affected by severe psychiatric conditions (such as psychotic mood disorders and schizophrenia), untreated endocrine diseases, and those that were admitted to the hospital in the previous three months or were unable to cooperate.

Blood samples were collected after overnight fasting in pyrogen-free tubes. EDTA was used as an anticoagulant for the quantification of cytokines, AGEs, sRAGEs, and esRAGEs. After centrifugation, plasma samples were stored at −80 °C until analysis.

The study was approved by the Ethics Committee of Fondazione IRCCS Policlinico (approval 347/2010), and all participants had to sign an informed written consent in agreement with the Helsinki Declaration.

### 2.2. Assessment of Depression and Cognitive Impairment

This assessment was performed just after the blood sample collection. The severity of depression was evaluated by the 30-item Geriatric Depression Scale (GDS; normal score ≤ 5), which was specifically developed for healthy and medically ill older adults with eventual mild/moderate cognitive impairment. The GDS resulted in 92% sensitivity and 89% specificity when evaluated against the diagnostic criteria of depression [34]. Of note, a score ≥ 10 is indicative of the presence of clinically significant depressive symptoms [35]. The GDS is a self-administered tool.

The Mini Mental State Examination (MMSE) is a widely used tool to conduct a first, albeit approximate, assessment of the cognitive functioning of the elderly [36]. Its score is adjusted for education, and a score above 25 (a maximum score is 30) is considered within normal limits [37].

The clock drawing test (CDT) (a normal value ≤ 1, 5 is the worst score) is another easy instrument to assess cognitive impairment in the elderly, especially spatial abilities and executive functions [38].

The MMSE and CDT were administered by trained clinicians.

### 2.3. Frailty Assessment

For the assessment of frailty, we used the frailty phenotype (FP) as proposed by Fried and colleagues [39]. Five components were used to define frailty: (1) involuntary weight loss ≥ 4.5 kg in 12 months; (2) exhaustion as feeling fatigued ≥4 days per week for more than 3 months; (3) weakness as handgrip strength < 16 kg in women and <27 kg in men; (4) slowness as a 4-m walk test speed < 0.8 m/s; (5) reduced physical activity with a score < 7 on a physical activity scale described elsewhere [40]. Patients with three or more impaired items were classified as frail.

### 2.4. Nutritional Status

Nutritional status was assessed by the determination of the individual malnutrition inflammation score (MIS) and the presence of protein energy wasting (PEW). The MIS was proposed by Kalantar-Zadeh et al. [41,42]. It is an adaptation of the Subjective Global Assessment (SGA) questionnaire, and it was associated with a worse prognosis in patients on haemodialysis [41,42,43], peritoneal dialysis [44,45], kidney transplantation [46], and in non-dialysed CKD patients [42]. The MIS transforms the SGA into a semiquantitative scoring system by adding some objective clinical and laboratory markers relevant to CKD. The MIS was validated against other nutritional/inflammatory biomarkers, and it is a composite score of 10 components, each with four severity levels: 0 (normal) to 3 (severely abnormal). A total score of 4–7 indicates a risk of malnutrition, and a score of ≥8 indicates malnutrition [47].

### 2.5. AGE Quantification

AGEs were quantified by a fluorometric method, as previously reported [48,49], using plasma samples. Briefly, 100 μL of each sample was added into a 96-well black microplate for fluorescence-based applications. Fluorescence intensity was measured at 414–445 nm after excitation at 365 nm using a fluorescence spectrophotometer (GloMax^®^-Multi Microplate Multi-mode Reader, Promega, Milan, Italy). The intensity of the fluorescence signal was expressed in arbitrary units (AUs). The AGE content was then normalized to the total protein content. The average inter- and intra-assay CV of fluorescent AGEs were calculated as 7.3% and 5.99%, respectively.

### 2.6. sRAGE, esRAGE, and cRAGE Quantification

The quantification of sRAGEs and their different forms—cRAGE and esRAGE—was carried out as previously described [50]. sRAGEs and esRAGEs were measured on plasma samples using two different ELISA kits: the R&D Systems kit (DY1145, Minneapolis, MN, USA) for sRAGEs and the B-Bridged International kit (K1009-1, Santa Clara, CA, USA) for esRAGEs. For sRAGE quantification, 100 μL of assay diluent and 50 μL of samples, controls, and standards were added to the 96-well microplate supplied by the kit. After a 2-h incubation and 4 washes, 200 μL of secondary conjugated antibody were added to each well. After another 2-h incubation and 4 washes, 200 μL of substrate solution were added to each well for 30 min, protected from light. After the addition of 50 μL of stop solution, absorbance was read at 450 nm with a wavelength correction set at 540 nm. For esRAGE quantification, 100 μL of assay diluent and 20 μL of samples, controls, and standards were added to the 96-well microplate supplied by the kit. After an overnight incubation at 4 °C, the plate was washed four times, and 100 μL of substrate solution were added to each well for 30 min. At the end of the incubation time, 100 μL of stop solution were added, and the absorbance was read at 450 nm with a wavelength correction set at 630 nm. For both sRAGEs and esRAGEs, standard curves were generated by using computer software capable of generating a four-parameter logistic (4-PL) curve fit (GraphPad Prism 9, Boston, MA, USA). cRAGE levels were obtained by subtracting esRAGEs from the total sRAGEs. The intra- and inter-assay coefficients of variation for the sRAGEs were 4.8–6.2% and 6.7–8.2%, respectively. For the esRAGE assay, they were 6.37 and 4.78–8.97%, respectively. Photometric measurements were performed using the GloMax^®^-Multi Microplate Multi-mode Reader (Promega, Milan, Italy).

### 2.7. Cytokine Quantification

The concentration of cytokines in the serum was determined in duplicate by using the following ELISA kits according to the instructions of the manufacturer: Quantikine ELISA Human CCL2/MCP-1 (Monocyte Chemoattractant Protein-1) Immunoassay DCP00, Human TNF-alpha (Tumor Necrosis Factor-alpha) ELISA Kit (Thermo Fisher Scientific, Monza, Italy), Quantikine HS ELISA Human Interleukin-6 (IL-6) Immunoassay HS600B (R&D Systems, Space, Milano, Italy), Human IL-10 ELISA Kit EHIL10 (Invitrogen, Thermo Fisher Scientific, Monza, Italy), Quantikine ELISA Human IL-12p70 Immunoassay D1200 (R&D Systems, Space, Milano, Italy), and Quantikine ELISA Human IL-17 Immunoassay D1700 kit (R&D Systems, Space, Milano, Italy). Each result curve had zero as the last standard value. The results were validated using the Quantikine Immunoassay Control Group 1–4 or 10 (R&D Systems, Space). A spectrophotometer (Xenius Safas, Monaco, Italy) was used to measure absorbance at 450 nm.

### 2.8. Statistical Analysis

We performed descriptive analyses of the total sample. The two groups, identified by the absence or presence of clinically significant depressive symptoms (GDS score < or ≥10), were compared by independent sample t tests for continuous variables and χ^2^ tests for qualitative ones with odds ratio (OR) and 95% confidence interval (CI) calculations where appropriate. Statistical significance was set at ≤0.05.

Statistically significant variables at univariate analyses were then inserted as predictors of a binary logistic regression model, with the presence or absence of depressive symptoms as a dependent variable. The reliability of the model was assessed by the Omnibus test.

The Statistical Package for Social Sciences (SPSS) for Windows (version 28.0) and GraphPad Prism 9 (Boston, MA, USA) were used as statistical programs.

## 3. Results

Descriptive analyses of the total sample and of the two groups identified by the presence of depressive disorders are reported in Table 1 (socio-demographic and clinical variables) and Table 2 (metabolic variables and systemic biomarkers). The sample consisted of 115 patients (80 males and 35 females) with a mean age of 79.7 ± 6.4 years and a mean eGFR of 24 ± 10 mL/min/1.73 m^2^. Forty-three patients (37.4%) presented depressive disorders corresponding to a GDS score ≥ 10. Only eight subjects (7.0%) had ongoing treatment with antidepressants. Patients with depressive symptoms (compared to their counterparts) were found to be more frequently women (χ^2^ = 13.94, *p* < 0.01, OR = 4.76 [95% CI: 2.04–11.10]), to be more frequently frail (χ^2^ = 11.51, *p* < 0.01), to have a higher MIS (t = 2.53, *p* = 0.01), to show lower CDT scores (t = 2.16, *p* = 0.03), and to present lower circulating MCP-1 levels (t = 2.99, *p* < 0.01) and lower AGE plasma levels (t = 2.20, *p* = 0.03). The sRAGEs, cRAGEs, and esRAGEs, as well as the other measured inflammatory markers (c-reactive protein—CRP, TNFα, IL-12p70, IL-17, and IL-10), did not differ between the two groups. Of note, no differences according to gender were detected regarding AGEs (t = 0.29, *p* = 0.77), sRAGEs (t = 0.98, *p* = 0.33), esRAGEs (t = 0.53, *p* = 0.60), cRAGEs (t = 1.23, *p* = 0.22), and AGEs/sRAGEs (t < 0.01, *p* = 1.00).

For research parameters not yet used in clinical practice (IL-10, IL-6, IL-17, IL-12p70, TNF-α, MCP-1, sRAGE, esRAGE, and cRAGE), the values of healthy controls were indicated in square brackets. They refer to values indicated by the diagnostic assay datasheets and/or cross-referencing to our previous data (sRAGE, esRAGE, and cRAGE). The data in square brackets include means and ranges, lower standards (IL-17, IL-12p70), lower detection limits (TNF-α), or medians and ranges (sRAGE, esRAGE, and cRAGE). All other data are expressed as means ± standard deviation.

The binary logistic regression model proved to be reliable (Omnibus test: *p* < 0.01), allowing a correct classification of 79.3% of cases. Patients with concomitant depressive disorders were more frequently of female gender (*p* < 0.01), as well as having lower MCP-1 (*p* < 0.01) and AGE circulating levels (*p* < 0.01) than their counterparts (Table 3; Figure 1).

## 4. Discussion

The first aspect to be discussed is that nearly 40% of our sample shows clinically significant depressive symptoms. This finding is not surprising because both advanced age [51] and the presence of a severe medical condition such as CKD [52,53] are risk factors for developing mood disorders. Despite the poor mental health of our sample, few patients had ongoing treatment with antidepressants at the time of evaluation. This phenomenon can be explained by different factors, including not administering psychopharmacotherapy to these patients for fear of adverse events [54] or the underestimation of symptoms by health professionals for the misleading idea that depression is somewhat physiological in elderly subjects [55].

With regard to clinical variables, female patients were more frequently affected by depressive disorders. This is a confirmation of previous data from the literature that report the prevalence of mood disturbances in women more than men, both in samples with primary psychiatric conditions [56] and in those with medical comorbidity [57]. The gender gap was explained by several factors, including the oscillation of estrogen levels throughout women’s lives [58] and a greater sensitivity of the HPA axis in women compared to men when exposed to stress, as in the case of serious medical conditions such as CKD [59].

Other results that are in agreement with the existing literature include the higher frequency of frailty and more severe malnourishment in CKD patients exhibiting depressive disorders [60]. Of note, these latter potentiate symptoms like loss of appetite, energy, and reduction in physical activity, which are already typical of CKD [60], thus increasing the associated disability [61]. Similarly, the dysregulation in depressive disorders of neuropeptides (e.g., ghrelin) that control appetite [62] enhances the metabolic alterations associated with poor nutrition in CKD [63]. In contrast, cognitive impairment seems to be more marked in subjects without than with depressive disorders. Different explanations can be considered with regard to this result: (1) other clinical factors (e.g., the presence of cerebrovascular disease) may have a greater impact on cognitive function [64]; (2) the CDT is not a sufficiently sensitive tool to capture the cognitive dysfunction of depressed subjects [65]; and (3) depressive disorders cause cognitive dysfunction in domains such as attention or memory that are less explored by the CDT [66].

With regard to systemic biomarkers, CKD patients affected by depressive disorders had lower plasma levels of MCP-1 and AGEs, as also confirmed by binary logistic regression models, while no differences were observed in the levels of the other evaluated inflammatory markers. These findings could be unexpected, as several data points indicate increased inflammation and oxidative stress in MDD [67,68]. Of note, previous studies found no significant differences in MCP-1 blood levels between patients and healthy controls [69], with even a negative correlation according to the severity of depression [70]. Different explanations can be considered with regard to this finding confirmed by the existing literature [71]: (1) MCP-1 may exert a protective role on mood disorders, balancing monoamine neurotransmission that results in altered depression [72]; (2) MCP-1 could induce a pro-inflammatory state and have a secondary role in the long term [73]; and (3) MCP-1 could accumulate in tissues where it exerts its chemoattractant and pro-inflammatory actions, resulting in a decrease in the plasma of patients with depression [74]. This latter hypothesis is supported by the fact that patients with CKD usually experience increased MCP-1 plasma levels [75] proportionally to renal damage [76], so this molecule could subsequently concentrate in brain tissue concomitantly with the development of depressive symptoms [77].

Finally, while the current literature indicates a negative association of AGEs with renal function [78,79], data regarding mental health are preliminary and conflicting. Elevated circulating AGEs would seem to be associated with poor cognitive function [80], while abnormal skin accumulation of AGEs would be detected in patients affected by schizophrenia and MDD [31]. This latter aspect could explain our finding of lower levels of plasma AGEs in CKD patients with depressive disorders compared to those without mood disturbances. Of note, elevated levels of some AGEs were found in the cerebrospinal fluid and the brain of patients affected by neurodegenerative disorders [81,82]. In addition to the locally produced AGEs, the increased availability of circulating AGEs may affect the integrity of the blood barrier in the brain, thus allowing AGEs to accumulate in the brain tissue and take part in pathways leading to mood disturbance [83]. A large-sample cross-sectional study confirmed that a higher skin accumulation of AGEs, but not of circulating ones, is associated with the onset and severity of depressive disorders [32]. An alternative explanation may be that the measurement of plasma AGEs in patients with depression is limited by the half-lives of these molecules and that the accumulation of these compounds in tissues is not reliable [84]. Finally, it might be that, since depression is associated with malnutrition in our study, depressed patients may have a reduced dietary intake that could translate to reduced AGE accumulation. In order to clarify whether these hypotheses may be true, future investigations on the role of AGEs in depressive disorders should also take into account the evaluation of dietary intake and the measurement of tissue AGEs.

Another important topic to address is how gender differences affect AGEs and the relationship between CKD and mental illness. It is still unclear how gender influences AGEs. While some studies indicated no difference between the two genders, others demonstrated that men accumulate more than women, or vice versa. Skin autofluorescence (AF) was the most commonly used assay for measuring AGE accumulation in tissues. In healthy Caucasian control subjects over a broad age range, gender had no influence on AF among non-smokers. Among smokers, female subjects had higher skin AF than male subjects [85]. The most important variables influencing AF were age and smoking. In healthy Chinese adults, AF was substantially associated with age and smoking, but no significant difference was identified between males and females [86]. In non-Caucasian individuals with type 2 diabetes mellitus, AF did not differ by gender [87]. Mook-Kanamori et al. [88] observed that gender has a significant effect on skin AF, with females having higher skin AF than men. The finding was consistent across all ethnicities and across both type 2 diabetes patients and controls. Immunohistochemical detection of Nε-carboxy-methyl-lysine, an AGE product, in the kidneys of two rat strains demonstrated greater accumulation in males than females [89]. Finally, a study about depressive symptoms found more frequent depressive symptoms in women than men, which in turn showed higher skin AGE levels. This aspect was explained by the fact that women were more frequently treated with antidepressants than men, and this is consistent with our findings where only eight subjects (three males and five females) were taking antidepressants, resulting in no gender differences in AGEs [90]. In contrast to some previous studies, we examined circulating AGEs, which is a more dynamic measure that accounts for both local synthesis and metabolism, as well as AGE intake. To summarize these observations, gender contribution cannot be ruled out, but a comparison between studies must be performed considering the type of AGEs assessed, the test utilized, and the clinical setting. In a clinical scenario like CKD, it is more likely that any potential difference between genders has been eliminated by the decreased kidney filtration, which is one of the main mechanisms inducing the accumulation of these molecules.

We recognize that our study has some limitations: (1) the lack of a follow-up as a result of the cross-sectional design of the study; (2) the lack of a control group with healthy individuals of the same age (of note, a largely lower prevalence of depression is reported in the general population [91]); (3) the fact that other factors may have contributed to the severity of the depressive symptoms; (4) the impossibility to establish a causal relationship between variables due to the study design; (5) the fact that the participants were selected from a specific CKD stage, potentially excluding those in earlier or more severe stages, although previous research failed to find a clear association between severity of renal disease and severity of depressive symptoms [9,92]; (6) the use of a self-rated scale like the GDS, which, despite being easy to administer, adds a subjective component to the assessment of depression and excludes the youngest part of patients, although CKD prevalence largely increases with age [93]; (7) the under-representation of females in our sample, although this feature was present in other samples [93], including an Italian one [94]; and (8) the potential impact of confounders such as lifestyle (e.g., physical exercise), emotional distress, and socio-economic status on the onset and severity of depressive symptoms.

However, we believe there are also some points of strength: we applied strict selection criteria that may have reduced the possible biases, and we accurately determined the associations of circulating AGEs and their receptors with depressive disorders in a not-insignificant number of older CKD patients.

## 5. Conclusions

In conclusion, depressive disorders more frequently characterize women affected by CKD and have a clinically negative impact, as shown by higher levels of frailty and more severe malnourishment. These aspects correspond to a decrease in MCP-1 and AGE plasma levels. Our findings come from data reflecting clinical practice, but they should be interpreted in light of the several study limitations. An in-depth study of AGEs in both CKD and mood disorders is of crucial importance given the roles of inflammatory aspects in both conditions. A strict collaboration between psychiatrists and nephrologists is needed to manage CKD patients with depressive disorders in order to ameliorate the prognosis of these subjects and prevent further deterioration of renal function. Further research is needed to confirm the results of the present study.

## Figures and Tables

**Figure 1 geriatrics-09-00099-f001:**
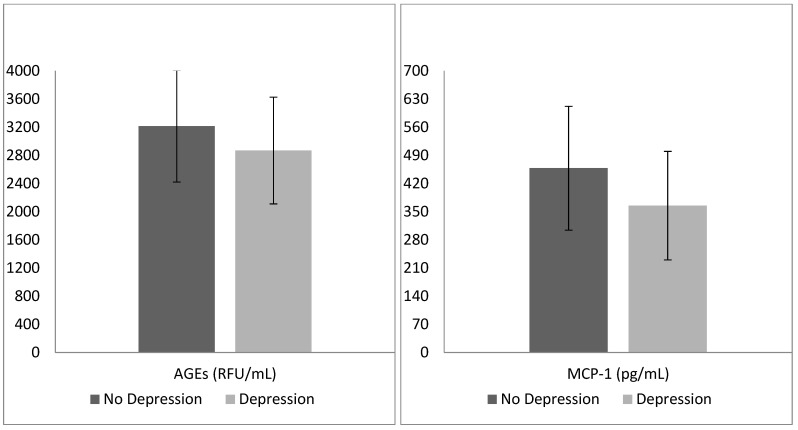
Mean advanced glycation end products (AGEs) and monocyte chemoattractant protein 1 (MCP-1) levels in the two groups identified by the presence of depressive disorders.

**Table 1 geriatrics-09-00099-t001:** Results of the socio-demographic and clinical variables of the total sample and of the two groups divided according to the presence of depressive disorders.

Variables		Total Sample	Depression −	Depression +	χ^2^ or t	OR (95% CI)	*p*
		N = 115	N = 72 (62.6%)	N = 43 (37.4%)			
**Sex**	*Male*	80 (69.6%)	59 (81.9%)	21 (48.8%)	13.94	4.76 (2.04–11.10)	**<0.01**
	*Female*	35 (30.4%)	13 (18.1%)	22 (51.2%)			
**Age (years)**		79.7 ± 6.4	79.2 ± 6.9	80.6 ± 5.3	1.12	N.A.	0.27
**Years of** **education**		10.8 ± 4.2	11.2 ± 4.4	10.1 ± 3.8	1.36	N.A.	0.18
**BMI (kg/m^2^)**		28.1 ± 4.8	28.1 ± 4.4	27.0 ± 5.5	0.14	N.A.	0.89
**Presence of a caregiver**	*Yes*	86 (74.8%)	55 (76.4%)	31 (72.1%)	0.26	0.80 (0.34–1.89)	0.61
	*No*	29 (25.2%)	17 (23.6%)	12 (27.9%)			
**Diabetes**	*Yes*	65 (56.5%)	43 (59.7%)	22 (51.2%)	0.8	0.71 (0.33–1.51)	0.37
	*No*	50 (43.5%)	29 (40.3%)	21 (48.8%)			
**Hypertension**	*Yes*	104 (90.4%)	66 (91.7%)	38 (88.4%)	0.34	0.69 (0.20–2.42)	0.56
	*No*	11 (9.6%)	6 (8.3%)	5 (11.6%)			
**Previous cardiovascular events**	*Yes*	63 (55.3%)	40 (56.3%)	23 (53.5%)	0.09	0.89 (0.42–1.91)	0.77
	*No*	51 (44.7%)	31 (43.7%)	20 (46.5%)			
**MIS**		6.24 ± 4.46	5.35 ± 3.28	7.74 ± 5.66	2.53	N.A.	**0.02**
**Frailty status**	*Yes*	52 (45.2%)	25 (34.7%)	27 (62.8%)	11.51	N.A.	**<0.01**
	*No*	15 (13.0%)	14 (19.5%)	1 (2.3%)			
	*Pre-frailty*	48 (41.8%)	33 (45.8%)	15 (34.9%)			
**MMSE**		25.8 ± 3.1	25.0 ± 3.1	25.9 ± 3.1	0.08	N.A.	0.94
**CDT**		3.3 ± 2.0	3.6 ± 1.9	2.8 ± 2.0	2.16	N.A.	**0.03**

BMI: body mass index; χ^2^ = chi square (qualitative variables); CDT: clock drawing test; CI: confidence interval; MIS: Malnutrition Inflammation Score; MMSE: Mini Mental State Examination; N.A.: not applicable; OR: odds ratio; *p*: *p* value; t: Student’s t value (continuous variables). Depression was defined by a Geriatric Depression Scale score ≥ 10. Statistically significant *p* values (≤ 0.05) are in bold. OR refers to patients with and without depressive disorders. Frequencies with percentages (into brackets) and means ± standard deviations were respectively reported for qualitative and continuous variables.

**Table 2 geriatrics-09-00099-t002:** Results of metabolic variables and the systemic biomarkers of the total sample and of the two groups divided according to the presence of depressive disorders.

Variables	Total Sample	Depression −	Depression +	t	*p*
	N = 115	N = 72 (62.6%)	N = 43 (37.4%)		
** *Biochemical variables* **					
eGFR (mL/min/1.73 m^2^)	24 ± 10	23 ± 10	25 ± 10	0.77	0.44
Creatinine clearance (mL/min)	27 ± 14	26 ± 13	28 ± 17	0.48	0.63
Albumin (g/dL)	4.0 ± 0.3	4.1 ± 0.4	4.0 ± 0.3	0.71	0.48
Total cholesterol (mg/dL)	166 ± 37	165 ± 36	170 ± 38	0.62	0.54
HDL (mg/dL)	53 ± 18	52 ± 18	54 ± 16	0.76	0.45
LDL (mg/dL)	113 ± 31	112 ± 29	114 ± 35	0.36	0.72
Vitamin D (ng/mL)	29 ± 17	28 ± 15	31 ± 19	0.95	0.35
** *Systemic inflammation* **					
CRP (mg/L) *	0.2 (0.1–0.4)	0.2 (0.1–0.5)	0.2 (0.1–0.4)	0.55	0.59
Leukocytes (cells/mm^3^)	6920 ± 1704	6951 ± 1856	6866 ± 1424	0.26	0.78
NLR	2.7 ± 1.4	2.8 ± 1.3	2.7± 1.4	0.37	0.71
IL-10 (pg/mL) [3.6; 0–14.1] *	1.9 (0.9–11.1)	1.9 (0.9–13.9)	1.8 (0.5–10.6)	0.21	0.83
Missing n = 10					
IL-6 (pg/mL) [43; 0–149]	3.9 ± 2.8	4.1 ± 2.7	3.6 ± 2.9	0.9	0.37
Missing n = 7					
IL-17 (pg/mL) [<31.3]	0.4 ± 1.1	0.2 ± 0.7	0.7 ± 1.6	1.57	0.13
Missing n = 9					
IL-12p70 (pg/mL) [<7.8]	1.7 ± 3.0	1.5 ± 2.3	1.9 ± 3.9	0.64	0.52
Missing n = 8					
TNF-α (pg/mL) [<2]	15.2 ± 8.2	15.0 ± 8.6	15.5 ± 7.6	0.32	0.75
Missing n = 8					
MCP-1 (pg/mL)	425 ± 154	458 ± 154	365 ± 135	2.99	**<0.01**
[423; 280.2–501.2]					
Missing n = 18					
** *Advanced glycation end products* **					
AGEs (RFU/mL)	3083 ± 794	3215 ± 793	2869 ± 759	2.2	**0.03**
Missing n = 10					
sRAGEs (pg/mL)	2346 ± 1305	2416 ± 1263	2232 ± 1380	0.7	0.48
[640; 365.5–1028]					
Missing n = 10					
esRAGEs (pg/mL)	663 ± 478	664 ± 470	662 ± 498	0.02	0.98
[338.7; 121.8–796.2]					
Missing n = 11					
cRAGEs (pg/mL)	1692 ± 961	1753 ± 969	1592 ± 953	0.82	0.41
[286.3; 2.23–564.7]					
Missing n = 11					
AGEs/sRAGEs	1.70 ± 0.99	1.69 ± 0.96	1.73 ± 1.04	0.18	0.86
Missing n = 10					

AGEs: advanced glycation end products; cRAGEs: membrane-cleaved receptors for AGEs; CRP: c-reactive protein; eGFR: estimated glomerular filtration rate; esRAGEs: endogenous secretory isoform of the receptor for AGEs; sRAGEs: total soluble receptors for AGEs; HDL: high-density lipoprotein; IL: interleukin; LDL: low-density lipoprotein; MCP-1: monocyte chemoattractant protein 1; NLR: neutrophil/lymphocyte ratio; TNF-α: tumor necrosis factor alpha; t: Student’s t value. Means ± standard deviations were reported. * For these variables, the median (25–75%) is reported.

**Table 3 geriatrics-09-00099-t003:** Summary of the statistics of the binary regression model.

Variables		B	SE	Wald	*p*	EXP(B)	95% CI for OR
**Sex**		2.38	0.79	9.05	**<0.01**	10.77	2.29–50.71
**Frailty**	**Pre-frail**	1.83	1.38	1.76	0.18	6.23	0.42–92.69
	**Frail**	1.09	1.23	0.79	0.38	2.98	0.27–33.24
**MIS**		0.16	0.1	2.71	0.1	1.17	0.97–1.42
**CDT score**		0.03	0.16	0.05	0.83	1.03	0.76–1.41
Missing n = 2							
**MCP-1 (pg/mL)**		−0.01	<0.01	8.85	**<0.01**	0.99	0.98–0.99
Missing n = 18							
**AGEs (RFU/mL)**		<−0.01	<0.01	8.87	**<0.01**	0.1	0.997–0.999
Missing = 10							

In this analysis, the dependent variable was the presence or absence of depressive disorders. AGEs: advanced glycation end products; B: regression coefficient; CDT: clock drawing test; CI: confidence interval; EXP(B): B exponential; MCP-1: monocyte chemoattractant protein 1; MIS: Malnutrition Inflammation Score; *p*: *p* value; RFU: relative fluorescent unit; SE: standard error of B; Wald: Wald statistics.

## Data Availability

The datasets generated and/or analyzed during the study are available from the corresponding author on request.

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
