# Peer review of "Clinical Factors and Biomarkers Associated with Depressive Disorders in Older Patients Affected by Chronic Kidney Disease (CKD): Does the Advanced Glycation End Products (AGEs)/RAGE (Receptor for AGEs) System Play Any Role?"

_geriatrics, 2024, doi:10.3390/geriatrics9040099_

Round 1
Reviewer 1 Report
Comments and Suggestions for Authors
This report examines whether AGEs are related to CKD and depression. It is a very interesting perspective. Since there are gender differences in the risk of depression and other conditions, it is desirable to measure the impact of AGEs taking this into account. It is also necessary to add a discussion on gender differences regarding AGEs. However, as this is a valuable report on CKD and depression, it would be good to complete it with a little more discussion.
Comments on the Quality of English LanguageThe level of English is acceptable.
Author Response
First of all, we would like to thank the reviewer for the interest in the manuscript and the useful suggestions aimed at improving the manuscript.
1) This report examines whether AGEs are related to CKD and depression. It is a very interesting perspective. Since there are gender differences in the risk of depression and other conditions, it is desirable to measure the impact of AGEs taking this into account.
Thanks for your observation. We performed a further analysis and no differences according to gender were detected regarding AGEs (t=0.29, p=0.77), sRAGEs, (t=0.98, p=0.33), esRAGEs (t=0.53, p=0.60), cRAGEs (t=1.23, p=0.22), AGEs/sRAGEs (t<0.01, p=1.00). A study about depressive symptoms found more frequent depressive symptoms in women than men that in turn showed higher skin AGEs levels. This aspect was explained by authors by the fact that women were more frequently treated with antidepressants than men, and this is consistent with our findings where only 8 subjects (3 males, 5 females, p > 0.05) were taking antidepressants, resulting no gender differences in AGEs (doi: 10.1016/j.jpsychores.2021.110488). We added all these considerations in the text with proper reference.
2) It is also necessary to add a discussion on gender differences regarding AGEs.
It is still unclear how gender influences AGEs. While some studies indicated no difference between the two genders, others demonstrated that men accumulate more than women, or vice versa. Skin autofluorescence (AF) was the most used assay for measuring AGEs accumulation in tissues. In healthy Caucasian control subjects over a broad age, gender had no influence on AF among nonsmokers. Among smokers, female subjects had higher skin AF than male (doi: 10.1089/dia.2009.0113). The most important variables influencing AF were age and smoking. In healthy Chinese adults, AF was substantially associated with age and smoking, but no significant difference was identified between males and females (doi: 10.1111/j.1464-5491.2010.03217.x). In non-Caucasian individuals with type 2 diabetes mellitus, AF did not differ by gender (doi: 10.1111/j.1464-5491.2011.03448.x). Mook-Kanamori et al. (doi: 10.4161/derm.26046) observed that gender has a significant effect on skin AF, with females having higher skin AF than men. The finding was consistent across all ethnicities and across both type 2 diabetes patients and controls. Immunohistochemical detection of CML, an AGEs product, in the kidneys of two rat strains demonstrated greater accumulation in males than females (doi: 10.1016/0024-3205(92)90230-m)
Differently from previous studies, we examined circulating AGEs, which is a more dynamic measure that accounts for both local synthesis and metabolism, as well as AGE intake. We also found no difference between the two genders. To summarize these observations, gender contribution cannot be ruled out, but comparison between studies must be performed considering the type of AGEs assessed, the test utilized, and the clinical setting. In a clinical scenario like CKD, it is more likely that any potential difference between genders has been eliminated by the decreased kidney filtration that is one the main mechanism inducing the accumulation of these molecules.
3)However, as this is a valuable report on CKD and depression, it would be good to complete it with a little more discussion.
Thanks for the appreciation and the suggestion: we expanded the discussion including all the points mentioned above.
Reviewer 2 Report
Comments and Suggestions for Authors
Paper merits for publication after minors corrections:
1- AGE quantification: Authors need to add in line 162 " AGE were mesured on plasma"
2- In table 2: CRP and IL10 have SD very high than mean, so they can be expressed by median (25%-75%)
Author Response
First of all, we would like to thank the reviewer for the interest in the manuscript and the useful suggestions aimed at improving the manuscript.
Paper merits for publication after minors corrections:
1) AGE quantification: Authors need to add in line 162 " AGE were measured on plasma"
We added this specification as requested.
2) In table 2: CRP and IL10 have SD very high than mean, so they can be expressed by median (25%-75%)
Thanks for your observation, we corrected accordingly
Reviewer 3 Report
Comments and Suggestions for Authors
Please see the attached file.

Comments on the Quality of English LanguageQuality of written English was Ok to me.
Author Response
First of all, we would like to thank the reviewer for the interest in the manuscript and the useful comments aimed at improving the manuscript.
1) There are some notable strengths in this study, i.e. it specifically targets older adults with CKD, examining factors associated with depressive disorders, particularly AGEs and sRAGE. With 115 participants, the study provides a substantial sample for analysis, enhancing statistical power. The use of the Geriatric Depression Scale (GDS) ensures a consistent and recognized tool for assessing depressive symptoms. Statistical methods including t-tests and logistic regression were employed to thoroughly analyze both clinical and biochemical factors. However, the study possesses a few weaknesses, like-being cross-sectional limits the ability to establish causal relationships between variables.
Thanks for your observation, we remarked the impossibility to establish a causal relationship between variables in the study limits.
2) Participants were selected from a specific CKD stage (3 to 5 not in dialysis), potentially excluding those in earlier or more severe stages.
We agree with your observation and we added this aspect as a limit of the study. We would like to point out that previous research (doi: 10.1016/j.jpeds.2015.09.040) including one by our group (doi: 10.1080/13548506.2018.1426868) failed to find a clear association between severity of renal disease and severity of depressive symptoms.
3) Reliance on self-reported depression scores could introduce variability due to subjective interpretation. Findings may not extend to younger CKD patients or those on dialysis, limiting broader applicability.
Thanks for your considerations. We added this aspect in the study limitations. The use of a self-rated scale like GDS, despite easy to administer, adds a subjective component to the assessment of depression and excludes the youngest part of patients, although CKD prevalence largely increases with age (doi: 10.1371/journal.pone.0158765).
4) Besides, studying advanced glycation-end products (AGEs) in CKD patients is crucial as AGEs are implicated in vascular damage and inflammation, both prevalent in CKD complications.
We agree with the reviewer and we remarked this aspect in the conclusions.
5) The mentioned study's methods for AGE quantification are comprehensive but could benefit from clearer reporting on calibration standards and quality control measures to enhance reproducibility. Additionally, specifying units for results and updating methodologies to reflect recent advancements would further strengthen the study's validity and comparability with other research. Understanding AGE involvement could lead to targeted interventions to mitigate vascular and inflammatory risks in advanced CKD, potentially improving patient outcomes.
The AGEs quantification is a homemade assay based on previously published methods (references 48 and 49). The test measures the fluorescence (365 nm excitation and 414-445 nm emission) of 100 uL-samples. Because no calibration curve was provided, the results are expressed as arbitrary units. We agree that the analysis may be less accurate than expected, but we assessed the intra- and inter-variability of the assay, which confirms that the results are reproducible. The average inter- and intra-assay CV of fluorescent AGEs were 7.3% and 5.99%, respectively. Protein concentrations were used to correct the detected signals for AGEs concentration. We thank the reviewer for his/her comments on AGEs quantification. This will encourage us to discontinue this method and improve novel methods of AGEs quantification. Lack of equipment and the cost of the novel and most updated processes are now some key difficulties we must deal with for the set-up of new AGEs quantification method and improve our future research. We agree that targeting AGEs and/or membrane RAGE could reduce some co-morbidities of CKD patients. To improve reader’s comprehension, we updated M&M as follows:
2.5. AGEs quantification
AGEs were quantified by a fluorometric method, as previously reported [48,49], using plasma samples. Briefly, 100 mL of each sample was added into a 96-well black microplate for fluorescence-based applications. Fluorescence intensity was measured at 414-445 nm after excitation at 365 nm using a fluorescence spectrophotometer (GloMax®-Multi Microplate Multi-mode Reader, Promega, Milan). The intensity of fluorescence signal was expressed as arbitrary units (AU). The AGEs content was then normalized to the total protein content. The average inter- and intra-assay CV of fluorescent AGEs were calculated and were 7.3% and 5.99%, respectively.
2.6. sRAGE, esRAGE and cRAGE quantification
The quantification of sRAGE and the different forms, cRAGE and esRAGE, was carried out as previously described [50]. sRAGE and esRAGE were measured on plasma samples using two different ELISA kits: R&D Systems kit (DY1145, Minneapolis, MN, USA) for sRAGE and B-Bridged International kit (K1009-1, Santa Clara, CA, USA) for esRAGE. For sRAGE quantification, 100 mL of assay diluent and 50 mL of samples, controls and standards were added to the 96-well microplate supplied by the kit. After 2-h incubation and 4 washes (300 mL/each), 200 mL of secondary conjugated antibody were added to each well. After 2-h incubation and 4 washes, 200 mL of substrate solution were added to each well for 30 minutes protected from light. After the addition of 50 mL of stop solution, absorbance was read at 450 nm with a wavelength correction set at 540 nm. For esRAGE,100 mL of assay diluent and 20 mL of samples, controls and standards were added to to the 96-well microplate supplied by the kit. After an overnight incubation at 4°, the plate was washed four times, 100 mL of substrate solution were added into each well for 30 minutes. At the end of the incubation time, 100 mL of stop solution were added and the plate and absorbance was read at 450 nm with a wavelength correction set at 630 nm. For both parameters, standard curves were generated by using a computer software capable of generating a four-parameter logistic (4-PL) curve fit (GraphPad Prism 9, Boston, MA). cRAGE levels were obtained by subtracting esRAGE from total sRAGE. The intra- and in-ter-assay coefficients of variation for sRAGE were 4.8-6.2% and 6.7-8.2%, respectively. For esRAGE assay they were 6.37 and 4.78-8.97%, respectively. Photometric measurements were performed using the GloMax®-Multi Microplate Multi-mode Reader (Promega).
6) Overall, while the study provides valuable insights into depressive disorders among older CKD patients and identifies potential biomarkers, cautious interpretation is necessary due to its limitations in study design and participant selection.
Thanks for your comment, we remarked this aspect in the conclusions.
Reviewer 4 Report
Comments and Suggestions for Authors
The manuscript by Buoli M and colleagues presents the results of a rather old study of co-occurrence of chronic kidney disease (CKD) and depression in patients recruited during Sep 2016–Mar 2018. The study tries to identify biomarkers for co-occurrence of CKD and depression, particularly focusing on advanced glycation-end products (AGE) and its receptors. The study builds on clinical scoring for depression using various methods and fluorescence and ELISA-based quantitation of various biomarkers in the recruited patients.
The subject area of the manuscript is of great value to the field of geriatrics, however, in the current form, the manuscript suffers from critical deficiencies as described below:
1. The biggest problem (also acknowledged by authors in the discussion section) is the lack of data from healthy controls in this study. It is therefore, hard to make sense of any data presented in this manuscript. For example, Table 1 shows certain % of male and female patients who have CKD and depression both. However, how significantly higher or lower the co-occurrence of depression with CKD is compared to age-matched healthy individuals can not be established in absence of data from non-CKD controls with and without depression. Similarly, data on biomarkers presented in Table 2 is meaningless in absence of similar data from healthy controls. No wonder, the manuscript results stand contradictory to the published results as far as biomarker levels in CKD are concerned.
2. The much smaller female subjects representation in the study makes the statistical inferences questionable. The manuscript suggests that female CKD patients tend to have higher incidences of depression compared to the male counterparts (Table 1). However, the female sample size is small (n=35), way smaller than male sample size (n=80). Due to uneven sample size, it is hard to trust the conclusion that women are more likely to have depression in CKD scenario compared to men.
3. There is no data on lifestyle, cultural background, socio-economic status, emotional well being and general mental health of subjects in this study. These factors may have significant contribution to depression experienced by an individual irrespective of their CKD condition. Hence, the data in the manuscript connecting depression with CKD can not be trusted.
4. There are no figures in the manuscript. Figures/charts are a better way to represent the data. It is easy to spot trend, differences, error bars, missing controls etc. in the data if presented in figures/charts. It is not clear why data was not plotted in the manuscript.
5. Methods have not been adequately described. For example, Section 2.2 mentions GDS, MMSE, CDT without describing how and when these were administered. it is not clear if all the samples were processed same day or on first-come first-serve basis. Details are missing for quantification of AGEs and various RAGEs. What was the sample—blood or serum used in these tests is not clear. Depending on what was used fluorescence read-out might have interference from several self-fluorescing bio molecules in blood. How was non-specific readout in fluorescence quantitation of biomarkers ruled out? What was the control is not clear.
Minor English editing needed.
Author Response
First of all, we would like to thank the reviewer for the interest in the manuscript and the precious comments aimed at improving the manuscript.
The biggest problem (also acknowledged by authors in the discussion section) is the lack of data from healthy controls in this study. It is therefore, hard to make sense of any data presented in this manuscript. For example, Table 1 shows certain % of male and female patients who have CKD and depression both. However, how significantly higher or lower the co-occurrence of depression with CKD is compared to age-matched healthy individuals can not be established in absence of data from non-CKD controls with and without depression. Similarly, data on biomarkers presented in Table 2 is meaningless in absence of similar data from healthy controls. No wonder, the manuscript results stand contradictory to the published results as far as biomarker levels in CKD are concerned.
Thanks for your considerations. The lack of a control group was remarked as a study limitation. With regard to the first comment, the prevalence of depression in general population corresponds to about 7% (doi: 10.1016/j.amepre.2022.05.014), largely lower than our sample of elderly patients affected by CKD. We added this consideration in the discussion.
With regard to the second aspect, a prior study by our research team (doi: 10.3390/jcm9113785) showed the levels of sRAGE, esRAGE, and cRAGE, but not AGEs, in a group of apparently healthy participants. The results were obtained using the same assays as in this study. Because these parameters, as well as inflammatory cytokines quantified in this research, have not yet been implemented in clinical practice and reference values are not available, we decided to improve the quality of table 2 by including the values of healthy controls as indicated by diagnostic assay datasheets or by cross-referencing to our previous data.
2) The much smaller female subjects representation in the study makes the statistical inferences questionable. The manuscript suggests that female CKD patients tend to have higher incidences of depression compared to the male counterparts (Table 1). However, the female sample size is small (n=35), way smaller than male sample size (n=80). Due to uneven sample size, it is hard to trust the conclusion that women are more likely to have depression in CKD scenario compared to men.
Thanks for your comment. The under-representation of female patients in our sample was added as a limit of the study, however it is in agreement of what reported in other samples (doi: 10.1371/journal.pone.0158765) including an Italian one (doi: 10.2215/CJN.02400310). On the other hand, it is well established in the literature that depressive disorders, even in the healthy population, are more frequent in women than in men (doi: 10.1016/S2215-0366(16)30263-2).
3) There is no data on lifestyle, cultural background, socio-economic status, emotional well being and general mental health of subjects in this study. These factors may have significant contribution to depression experienced by an individual irrespective of their CKD condition. Hence, the data in the manuscript connecting depression with CKD can not be trusted.
Thanks for your observation and we added all the cited variables as potential confounders in the limits of the present study. However this study already involved several variables and the addition of others would have amplified the statistical error due to multiple tests (multiplicity). We remarked the fact that our results should be cautiously interpreted in the light of several limits. With regard to general mental health, the presence of severe psychiatric conditions (such as psychotic mood disorders and schizophrenia) was an exclusion criterion.
4) There are no figures in the manuscript. Figures/charts are a better way to represent the data. It is easy to spot trend, differences, error bars, missing controls etc. in the data if presented in figures/charts. It is not clear why data was not plotted in the manuscript.
Thanks for your suggestion. We added figure 1 to address your request.
5) Methods have not been adequately described. For example, Section 2.2 mentions GDS, MMSE, CDT without describing how and when these were administered. it is not clear if all the samples were processed same day or on first-come first-serve basis. Details are missing for quantification of AGEs and various RAGEs. What was the sample—blood or serum used in these tests is not clear. Depending on what was used fluorescence read-out might have interference from several self-fluorescing bio molecules in blood. How was non-specific readout in fluorescence quantitation of biomarkers ruled out? What was the control is not clear.
With regard to rating scales we added in the text what you requested. Depression evaluation and cognitive assessment were performed just after the blood collection. The Geriatric Depression Scale is a self-administered tool. The cognitive tests were administered by trained clinicians.
Blood samples were collected in EDTA-containing tubes, immediately processed and plasma was stored at -80°C until analyses, which were performed in the same analytical session for each marker. AGEs quantification is a homemade assay based on previously published methods (references 48 and 49). The test measures the fluorescence (365 nm excitation and 414-445 nm emission) of 100 uL-samples. Because no calibration curve was provided, the results are expressed as arbitrary units. We agree that the analysis may be less accurate than expected, but we assessed the intra- and inter-variability of the assay, which confirms that the results are reproducible. The average inter- and intra-assay CV of fluorescent AGEs were 7.3% and 5.99%, respectively. Protein concentrations were used to correct the detected signals for AGEs concentration. We thank the reviewer for his/her comments on AGEs quantification. This will encourage us to discontinue this method and set up novel methods of AGEs quantification for improving accuracy. We also updated the M&M as follows:
In section 2.1: “Blood samples were collected after overnight fasting into pyrogen-free tubes. EDTA was used as anticoagulant for the quantification of cytokines, AGEs, sRAGE and esRAGE. After centrifugation, plasma samples were stored at -80° C until analyses.”
2.5. AGEs quantification
AGEs were quantified by a fluorometric method, as previously reported [48,49], using plasma samples. Briefly, 100 mL of each sample was added into a 96-well black microplate for fluorescence-based applications. Fluorescence intensity was measured at 414-445 nm after excitation at 365 nm using a fluorescence spectrophotometer (GloMax®-Multi Microplate Multi-mode Reader, Promega, Milan). The intensity of fluorescence signal was expressed as arbitrary units (AU). The AGEs content was then normalized to the total protein content. The average inter- and intra-assay CV of fluorescent AGEs were calculated and were 7.3% and 5.99%, respectively.
2.6. sRAGE, esRAGE and cRAGE quantification
The quantification of sRAGE and the different forms, cRAGE and esRAGE, was carried out as previously described [50]. sRAGE and esRAGE were measured on plasma samples using two different ELISA kits: R&D Systems kit (DY1145, Minneapolis, MN, USA) for sRAGE and B-Bridged International kit (K1009-1, Santa Clara, CA, USA) for esRAGE. For sRAGE quantification, 100 mL of assay diluent and 50 mL of samples, controls and standards were added to the 96-well microplate supplied by the kit. After 2-h incubation and 4 washes, 200 mL of secondary conjugated antibody were added to each well. After 2-h incubation and 4 washes, 200 mL of substrate solution were added to each well for 30 minutes, protected from light. After the addition of 50 mL of stop solution, absorbance was read at 450 nm with a wavelength correction set at 540 nm. For esRAGE,100 mL of assay diluent and 20 mL of samples, controls and standards were added to the 96-well microplate supplied by the kit. After an overnight incubation at 4°, the plate was washed four times and 100 mL of substrate solution were added into each well for 30 minutes. At the end of the incubation time, 100 mL of stop solution were added and the plate and absorbance was read at 450 nm with a wavelength correction set at 630 nm. For both sRAGE and esRAGE, standard curves were generated by using a computer software capable of generating a four-parameter logistic (4-PL) curve fit (GraphPad Prism 9, Boston, MA). cRAGE levels were obtained by subtracting esRAGE from total sRAGE. The intra- and inter-assay coefficients of variation for sRAGE were 4.8-6.2% and 6.7-8.2%, respectively. For esRAGE assay they were 6.37 and 4.78-8.97%, respectively. Photometric measurements were performed using the GloMax®-Multi Microplate Multi-mode Reader (Promega).
Round 2
Reviewer 4 Report
Comments and Suggestions for Authors
The revised version of the manuscript and accompanying author's rebuttal adequately address all the comments raised by this reviewer. I appreciate authors for their time and effort revising the manuscript.